# For Universal Multiclass Online Learning, Bandit Feedback and Full Supervision are Equivalent

**Steve Hanneke**                                            STEVE.HANNEKE@GMAIL.COM
*Department of Computer Science, Purdue University, USA*

**Amirreza Shaeiri**                                    AMIRREZA.SHAEIRI@GMAIL.COM
*Department of Computer Science, Purdue University, USA*

**Hongao Wang**                                              WANG5270@PURDUE.EDU
*Department of Computer Science, Purdue University, USA*

**Editors:** Gautam Kamath and Po-Ling Loh

## Abstract

We study the problem of multiclass online learning under *bandit feedback* within the framework of *universal learning* (Bousquet, Hanneke, Moran, van Handel, and Yehudayoff, 2021).

In multiclass online learning under bandit feedback, it is well known that no concept class $\mathcal{C}$ is *uniformly* learnable when the effective label space is unbounded, or in other words, no online learner guarantees a finite bound on the expected number of mistakes holding uniformly over all realizable data sequences. In contrast, surprisingly, we show that in the case of *universal* learnability of concept classes $\mathcal{C}$, there is an exact equivalence between multiclass online learnability under bandit feedback and full supervision, both in the realizable and agnostic settings.

More specifically, our first main contribution is a theory that establishes an inherent dichotomy in multiclass online learning under bandit feedback within the realizable setting. In particular, for any concept class $\mathcal{C}$ even when the effective label space is unbounded, we have: (1) If $\mathcal{C}$ does not have an infinite multiclass Littlestone tree, then there is a deterministic online learner that makes only finitely many mistakes against any realizable adversary, crucially without placing a uniform bound on the number of mistakes. (2) If $\mathcal{C}$ has an infinite multiclass Littlestone tree, then there is a strategy for the realizable adversary that forces any online learner, including randomized, to make linear expected number of mistakes. Furthermore, our second main contribution reveals a similar trend in the agnostic setting.

**Keywords:** Online Learning, Universal Learning, Bandit Feedback, Multiclass Learning

## 1. Introduction

Consider a scenario where an advertising service endeavors to present advertisements from a predetermined set of ads to visitors worldwide by leveraging their data. Each user exhibits a propensity to engage with only one specific advertisement within that set. When an advertisement is suggested by the service, users respond either by clicking on it, thereby indicating engagement, or by ignoring it, demonstrating a lack of engagement. In this context, a reasonable objective for the company is to reduce the number of occurrences in which users do not engage with an ad over time. How can we rigorously formalize and address this scenario with minimal assumptions?

The aforementioned example can be formulated within the framework known as *Multiclass Online Learning under Bandit Feedback*. Broadly speaking, this framework is defined as a sequential prediction problem wherein probabilistic assumptions regarding the data-generating process are intentionally omitted. More precisely, during each round, an adversary first selects an instance (e.g., an image) from an instance space. Subsequently, the learner is tasked with predicting a label (e.g., an image category) from a label space for the received instance. Upon the learner's prediction, the adversary only reveals whether the prediction is correct. Importantly, this setup differs from a full supervision/information setup, where the true label is revealed by the adversary at the end of each round. Furthermore, the main quantity of interest in this framework is the well-known notion of the number of mistakes made by the learner over time. It is important to note that for the sake of simplicity, we have mentioned the formulation specifically for deterministic learners.

To derive meaningful results, this work considers a well-established notion of a concept class containing functions from the instance space to the label space. In the realizable setting, we assume that the sequence played by the adversary is consistent with one of the concepts in the concept class [1]. Moreover, in this setting, we focus exactly on the number of mistakes made by the learner as an objective. Conversely, in the agnostic setting, we make no assumptions about the sequence played by the adversary. In this setting, instead of considering the number of mistakes made by the learner as an objective, we compare the learner's number of mistakes to that of the best concept within the concept class [2], a well-known notion called regret. We briefly note that if the learner's predictions are randomized, we focus on the expected value of the mentioned objectives. For simplicity, in this part of the introduction, we primarily focus on the realizable setting.

Prior research on this problem has only considered the *uniform* learning framework. In this framework, in the realizable setting, the goal is to design online learning algorithms that achieve a *uniformly bounded* expected number of mistakes against any realizable adversary. This line of study was initiated by the work of Daniely, Sabato, Ben-David, and Shalev-Shwartz (2011), where they introduced a combinatorial complexity parameter called bandit Littlestone dimension, which characterizes the best achievable worst-case number of mistakes in the realizable setting. Subsequently, Daniely and Helbertal (2013); Long (2017); Geneson (2021); Raman, Raman, Subedi, and Tewari (2023); Hanneke and Yang (2023) continued this line of work in both the realizable and the

---

1. Technically, in the universal framework, each *prefix* of the sequence played by the adversary must be consistent with some concept in the concept class. We refer the reader to section 2 and section 3 for more details.
2. Technically, in the universal framework, we compare the learner's number of mistakes to that of the best concept within the *closed version* of the concept class over a sequence played by the adversary. We refer the reader to section 2 and section 4 for more details.

agnostic settings. If the number of labels is infinite, particularly if there exists at least one instance in the instance space for which the concepts in the concept class can take on an infinite number of distinct labels, no online learner guarantees to achieve a finite bound on the number of mistakes holding uniformly against any realizable adversary. To illustrate this, consider a scenario in which the adversary plays that specific instance in every round. Any prediction of the learner can be incorrect, while a concept in the concept class that is consistent with all previous feedback still exists. The mentioned argument can be easily extended to randomized online learners. Consequently, in multiclass online learning under bandit feedback within the uniform framework, no concept class is learnable when the effective label space is unbounded. This includes elementary and natural classes such as an infinite class of constant functions on natural numbers, which is online learnable with at most one mistake in the full supervision setup.

In response to the above theoretical limitation, a potential approach is considering sequence-dependent mistake bounds. Thus, in this work, we focus on an arguably more realistic framework of *universal* learning Bousquet, Hanneke, Moran, van Handel, and Yehudayoff (2021). In this framework, in the realizable setting, the goal is to design learning algorithms that achieve a finite expected number of mistakes against any realizable adversary, crucially *without placing a uniform bound* on the number of mistakes. Let us examine an instance of multiclass online learning under the bandit feedback problem involving an instance space with only one element, a label space as natural numbers, and a concept class encompassing all possible functions from the given instance space to the specified label space. Now, consider the following learning algorithm: In each round $i \in \mathbb{N}$, the algorithm predicts the label $i$ unless it receives feedback indicating that the prediction is correct, after which it consistently predicts the correct label. Clearly, this algorithm guarantees a finite number of mistakes for any realizable sequence. As a result, the universal learning framework may effectively overcome the aforementioned theoretical challenge. Perhaps surprisingly, our results not only tackle the mentioned theoretical limitation, but also reveal an unexpected exact equivalence between multiclass online learning under bandit feedback and full supervision within the universal learning framework, in both the realizable and the agnostic settings. Therefore, shockingly, we can learn significantly broader concept classes in multiclass online learning under bandit feedback within the universal framework compared to multiclass online learning under full supervision within the uniform framework.

## 1.1. Related Work and Backgrounds

**Online Learning.** One of the most fundamental and classical topics in statistical learning theory is online learning, having been rigorously studied for over 35 years within the computer science community. The seminal work of Littlestone (1988) initiated this line of research by introducing the problem of online binary classification under full supervision. Littlestone's research demonstrated that a combinatorial complexity parameter, known as the Littlestone dimension, characterizes the best achievable worst-case number of mistakes in the online binary classification framework in the realizable setting. Subsequently, Ben-David, Pál, and Shalev-Shwartz (2009) showed that the Littlestone dimension continues to characterize the learnability in the agnostic variant of the same framework. Since these pioneering contributions, online learning has been extensively explored across various settings, including online learning under bandit feedback. It is also linked to a broad list of problems, such as differential privacy, as evidenced by studies conducted by Alon, Livni, Malliaris,

and Moran (2019); Bun, Livni, and Moran (2020); Alon, Bun, Livni, Malliaris, and Moran (2022); Fioravanti, Hanneke, Moran, Schefler, and Tsubari (2024). Moreover, given its fundamental nature, it is unsurprising that online learning has garnered numerous practical real-world applications.

**Bandit Feedback.** Bandit setting occupies a central position in statistical decision-making. The conceptual foundation for bandit problems was initially laid by Thompson (1933). Subsequently, the topic was popularized by the mathematician and statistician Herbert Robbins, beginning with his seminal work Robbins (1952), which introduced the multi-armed bandit problem. More recently, bandit scenarios have received a lot of attention, starting from the seminal works of Auer, Cesa-Bianchi, and Fischer (2002a); Auer, Cesa-Bianchi, Freund, and Schapire (2002b). We suggest that the readers see the recent work of Foster, Kakade, Qian, and Rakhlin (2021) for a more detailed review of the literature.

**Universal Learning.** The notion of universal consistency has been extensively studied over a long period, originating from works within the statistics community. However, the notion of universal learnability considered in the current article was recently introduced and popularized by the seminal work of Bousquet, Hanneke, Moran, van Handel, and Yehudayoff (2021). Although the literature on universal learning under full supervision is extensive, there are no known theoretical results for the case of bandit setting. The studies most closely related to our work are the recent contributions by Blanchard, Hanneke, and Jaillet (2022) and Blanchard, Hanneke, and Jaillet (2023). Their focus, however, is on the universal consistency across different families of processes, crucially without any notable restriction on the functions being learned. Thus, our work significantly diverges because it focuses on universal learnability when we have the notion of a concept class. For a more comprehensive explanation of the distinctions between these frameworks, we recommend referring to the work of Hanneke (2021).

**Multiclass Learning.** A substantial volume of theoretical research, driven by practical applications, has been conducted on multiclass learning, including studies by Natarajan and Tadepalli (1988); Natarajan (1989); Ben-David, Cesa-Bianchi, and Long (1992); Haussler and Long (1995); Rubinstein, Bartlett, and Rubinstein (2006); Daniely, Sabato, Ben-David, and Shalev-Shwartz (2011); Daniely, Sabato, and Shwartz (2012); Daniely and Shalev-Shwartz (2014); Brukhim, Hazan, Moran, Mukherjee, and Schapire (2021). Despite this, a combinatorial characterization of multiclass learning, when we have an infinite label space, within Valiant's PAC learning framework remained unresolved until recently. The seminal work of Brukhim, Carmon, Dinur, Moran, and Yehudayoff (2022) building upon the work of Daniely and Shalev-Shwartz (2014) provided this characterization, which also applies to the agnostic variant of the framework David, Moran, and Yehudayoff (2016). Additionally, Kalavasis, Velegkas, and Karbasi (2022); Hanneke, Moran, Raman, Subedi, and Tewari (2023a) have studied this problem within the universal learning framework. For multiclass online learning with an infinite label space framework, Daniely, Sabato, Ben-David, and Shalev-Shwartz (2011) characterized the realizable setting by introducing the multiclass Littlestone dimension. Subsequently, this result was recently extended by Hanneke, Moran, Raman, Subedi, and Tewari (2023a) to the agnostic setting.
Notably, motivations for studying multiclass learnability with an infinite label space include ensur-

ing that guarantees are not dependent on the number of labels, gaining insight by studying infinities, and advancing understanding in real-valued regression Attias, Hanneke, Kalavasis, Karbasi, and Velegkas (2023). Practically, tasks such as image object recognition and next-word prediction involve a large number of classes.

## 1.2. Overview of the Main Results

In the following subsection, we provide a detailed summary of the key results and findings presented in our paper.

### 1.2.1. MULTICLASS ONLINE LEARNING UNDER BANDIT FEEDBACK FRAMEWORK

We consider a sequential game between the learner and an adversary. At each round $t \in \mathbb{N}$, the adversary chooses an instance $X_t$ from an arbitrarily non-empty instance space $\mathcal{X}$ and a label $Y_t$ from a possibly countably infinite non-empty label space $\mathcal{Y}$ [3]; then, reveals $X_t$ to the learner. Following this, the learner predicts a label $\hat{Y}_t$ from $\mathcal{Y}$, which can possibly be randomized. Subsequently, the learner, instead of observing the true label $Y_t$, receives feedback $\mathbb{I}\left[\hat{Y}_t \neq Y_t\right]$, indicating whether the prediction is correct. Before proceeding, following the well-established frameworks in learning theory, we define a concept class $\mathcal{C}$ as a set of functions from $\mathcal{X}$ to $\mathcal{Y}$, which is known to the learner before starting the game. For more details, see section 2.

### 1.2.2. REALIZABLE SETTING

In the realizable setting, we assume that each prefix of the sequence $\{(X_t, Y_t)\}_{t=1}^{\infty}$, played by the adversary, is consistent with at least one concept in $\mathcal{C}$. See section 3 for more details.

Here, our objective is to minimize the well-known notion of the expected number of mistakes over time. We say that the concept class $\mathcal{C}$ is *learnable* in the multiclass online learning under bandit feedback framework in the realizable setting, if there is an online learning algorithm that makes only finitely many mistakes in expectation on any realizable sequence played by the adversary, crucially without imposing a uniform bound on the expected number of mistakes. The main result of this part demonstrates that this criterion for learnability is fully characterized by the non-existence of infinite multiclass Littlestone trees. This notion is closely related to the multiclass Littlestone dimension Daniely, Sabato, Ben-David, and Shalev-Shwartz (2011). However, it is essential to note that having an infinite multiclass Littlestone tree is different from having an unbounded multiclass Littlestone dimension; the latter can occur due to the presence of finite Littlestone trees of arbitrarily large depth, which does not necessarily imply the existence of any single tree of infinite depth. See section 3 and Bousquet, Hanneke, Moran, van Handel, and Yehudayoff (2021) for more details. Formally, we have the following surprising theorem.

---

3. Following the work of Hanneke et al. (2023a), if we consider randomized online learning algorithms, the associated $\sigma$-algebra is of little consequence, except that singleton sets $\{y\}$ should be measurable.

**Theorem 1** *Let $\mathcal{C} \subseteq \mathcal{Y}^{\mathcal{X}}$ be a concept class. Then, we have the following dichotomy:*

- *If $\mathcal{C}$ does not have an infinite multiclass Littlestone tree, then there exists a deterministic online learning algorithm that makes only finitely many mistakes against any realizable adversary.*

- *If $\mathcal{C}$ has an infinite multiclass Littlestone tree, then there is a strategy for the realizable adversary that forces any online learning algorithm, including randomized, to make a linear expected number of mistakes.*

*In particular, $\mathcal{C}$ is learnable in the multiclass online learning under bandit feedback framework in the realizable setting if and only if it has no infinite Littlestone tree.*

The work of Hanneke, Moran, and Zhang (2023b) demonstrates exactly the same combinatorial characterization for the universal multiclass online learnability under full supervision in the realizable setting (see Theorem 18 in Hanneke, Moran, and Zhang (2023b)). Therefore, our result reveals an unexpected exact equivalence between multiclass online learning under bandit feedback and full supervision within the universal learning framework in the realizable setting. Formally, we have the following corollary.

**Corollary 2** *Let $\mathcal{C} \subseteq \mathcal{Y}^{\mathcal{X}}$ be a concept class. Then, $\mathcal{C}$ is learnable in the multiclass online learning under bandit feedback framework in the realizable setting if and only if $\mathcal{C}$ is learnable in the multiclass online learning framework in the realizable setting.*

### 1.2.3. AGNOSTIC SETTING

In the agnostic setting, we make no assumptions about the sequence $\{(X_t, Y_t)\}_{t=1}^{\infty}$, played by the adversary. See section 4 for more details.

Here, our attention shifts to minimizing the well-known notion of expected regret, which quantifies the difference between the expected number of mistakes made by the learning algorithm and that made by the best concept within the closed version of the concept class over a sequence played by the adversary. Notably, we work with the closed version of the concept class to ensure that the realizable setting is a special case of the agnostic setting. We say that the concept class $\mathcal{C}$ is *learnable* in the multiclass online learning under bandit feedback framework in the agnostic setting, if there is an online learning algorithm that has a sub-linear expected regret, on any sequence played by the adversary, crucially without placing a uniform bound on the expected regret. The main result of this part establishes that this criterion for learnability is also fully characterized by the non-existence of infinite multiclass Littlestone trees. Formally, we present the following surprising theorem.

**Theorem 3** *Let $\mathcal{C} \subseteq \mathcal{Y}^{\mathcal{X}}$ be a concept class. Then, we have the following dichotomy:*

- *If $\mathcal{C}$ does not have an infinite multiclass Littlestone tree, then there exists an online learning algorithm that has $o(T)$ expected regret against any adversary.*

- *If $\mathcal{C}$ has an infinite multiclass Littlestone tree, then there is a strategy for the adversary that forces any online learning algorithm to have a linear expected regret.*

*In particular, $\mathcal{C}$ is learnable in the multiclass online learning under bandit feedback framework in the agnostic setting if and only if it has no infinite Littlestone tree.*

Our result in this part also reveals an unexpected exact equivalence between multiclass online learning under bandit feedback and full supervision within the framework of universal learning in the agnostic setting. Notably, before our work, even the characterization of universal multiclass online learnability under full supervision in the agnostic setting had not yet been established. Formally, we have the following corollary.

**Corollary 4** *Let $\mathcal{C} \subseteq \mathcal{Y}^{\mathcal{X}}$ be a concept class. Then, $\mathcal{C}$ is learnable in the multiclass online learning under bandit feedback framework in the agnostic setting if and only if $\mathcal{C}$ is learnable in the multiclass online learning framework in the agnostic setting.*

### 1.3. Overview of the Techniques

In the following subsection, we provide a summary of the key techniques in the proof of the above two theorems 1 3.

#### 1.3.1. REALIZABLE SETTING

In contrast to the algorithm and the proof of the discussed theorem for universal multiclass online learnability under full supervision in the realizable setting in the work of Hanneke, Moran, and Zhang (2023b), our algorithm and the proof behind the realizable theorem 1 are considerably more involved. Specifically, our algorithm is built upon the celebrated weighted majority algorithm Littlestone and Warmuth (1994); Long (2017); Hanneke, Livni, and Moran (2021); Hanneke and Yang (2023). The primary innovation in our approach lies in the introduction of certain pre-specified weights corresponding to each $y \in \mathcal{Y}$, alongside the replacement of the Standard Optimal Algorithm (SOA) with the ordinal SOA from Hanneke, Moran, and Zhang (2023b). These weights form a non-uniform probability distribution over $\mathcal{Y}$, with a non-zero probability mass assigned to each $y \in \mathcal{Y}$. Having these two ideas at hand, the proof of the desired result is technically similar to that of Long (2017). Notably, we also provide a simpler proof of this theorem. For more details, see section 3.

#### 1.3.2. AGNOSTIC SETTING

Before discussing our techniques for proving the result in the agnostic setting, we first make the definitions precise, which can also be regarded as one of our contributions, as it involves new thoughts. In adversarial online learning frameworks in the agnostic setting, in line with prior work, we should

have the flexibility to choose the sequence played by the adversary adaptively and without restriction. Therefore, in the multiclass online learning under bandit feedback framework in the agnostic setting, the adversary plays instances based on $\{X_t\}_{t=1}^{\infty}$ and provides feedback based on $\{Y_t\}_{t=1}^{\infty}$, both of which may be chosen adversarially and adaptively without any restriction. To put it another way, we make no assumptions about the sequence $\{(X_t, Y_t)\}_{t=1}^{\infty}$. Furthermore, following all previous papers on adversarial online learning frameworks in the agnostic setting, we are interested in the notion of regret, necessitating the existence of something to compete with. However, if we standardly compare the algorithm's number of mistakes to the number of mistakes made by the best concept within the concept class over the sequence played by the adversary, the best concept within the concept class can potentially change over time, which contrasts with the nature of the universal learning framework, where the learning problem should not change its past. One way to handle this is to consider a fixed concept within the concept class to compete with, but the more natural way to handle seems to compete against realizable sequences from a universal perspective. As a result, in the spirit of the universal learning framework, in the multiclass online learning under bandit feedback framework in the agnostic setting, basically, we consider a sequence of three tuples $\{(X_t, Y_t, Y_t^*)\}_{t=1}^{\infty}$, where $\{Y_t^*\}_{t=1}^{\infty}$ may also be adversarially chosen, but with the restriction that each prefix of the sequence $\{(X_t, Y_t^*)\}_{t=1}^{\infty}$ must be consistent with at least one concept in $\mathcal{C}$. Specifically, $\{Y_t^*\}_{t=1}^{\infty}$ represents the sequence against which we aim to compete. Notably, this formalization ensures that the realizable setting becomes a special case of the agnostic setting when $\{Y_t\}_{t=1}^{\infty}$ and $\{Y_t^*\}_{t=1}^{\infty}$ are equivalent. Importantly, the learner never sees $\{Y_t^*\}_{t=1}^{\infty}$. For more details, see section 2.

Subsequently, we shift our focus to the techniques behind the proof of the agnostic theorem 3. In the agnostic setting, our algorithm is designed based on two main components: (1) The construction of experts by utilizing the realizable algorithm from section 3, in a manner similar to prior works in online learning, beginning with the work of Ben-David et al. (2009). (2) The application of the EXPINF algorithm proposed in Blanchard, Hanneke, and Jaillet (2022), instead of the EXP4 algorithm employed in the work of Daniely and Helbertal (2013) within the uniform setting. Having these two ideas at hand, with a bit of effort, we can prove the desired result. For more details, see section 4.

### 1.4. Organization

The rest of our paper is organized as follows. In section 2, we formally set notations and definitions. Subsequently, in section 3, we present our results for the realizable setting. Following this, in section 4, we extend our results to the agnostic setting. Eventually, in section 5, we conclude our manuscript and present some recommended directions for future work.

## 2. Notations, Definitions, and Preliminaries

In this section, we present the model setting and formal definitions.

**Model Setting.** Let $\mathcal{X}$ be a non-empty *instance space* and $\mathcal{Y}$ a non-empty but countable *label space*. And in this paper, we focus on learning under the 0-1 loss: that is, $(y, y') \mapsto \mathbb{I}[y \neq y']$

defined on $\mathcal{Y} \times \mathcal{Y}$, where $\mathbb{I}[\cdot]$ is the indicator function. A data sequence $\{X_t\}_{t \in \mathbb{N}}$ is a sequence of $\mathcal{X}$-valued elements. A label sequence $\{Y_t\}_{t \in \mathbb{N}}$ is a sequence of $\mathcal{Y}$-valued elements. The concept class $\mathcal{C} \subseteq \mathcal{Y}^{\mathcal{X}}$ is a non-empty set of functions from $\mathcal{X}$ to $\mathcal{Y}$.

**Full Supervision.** Multiclass online classification with *full supervision* is a sequential game. There are two players in the game. They are the learner $L$ and the adversary $A$. In each round $t$, the adversary chooses an instance $X_t$ and the true label $Y_t$ adaptively and show $X_t$ to the learner and then the learner predicts the label $\hat{Y}_t = \hat{h}_{t-1}(X_t)$ based on the entire history up to the current round, $(X_{<t}, Y_{<t}) = \{(X_i, Y_i)\}_{i<t}$. Thus, this prediction can also be expressed as $f_t(X_{<t}, Y_{<t}, X_t)$. After the prediction, the adversary will reveal the true label $Y_t$, which can be used to inform the future prediction, to the learner. In the realizable setting, the true label chosen by the adversary should satisfy the constraint that for any $t \in \mathbb{N}$, there is a concept $c \in \mathcal{C}$, such that for any $t' < t$, $Y_{t'} = c(X_{t'})$. We will call that type of adversary as realizable adversary. We use the number of mistakes to describe whether a concept class $\mathcal{C}$ is learnable in the realizable setting. Formally, the number of mistakes is defined as follows.

**Definition 5 (Number of mistakes)** *For an online learning rule $\hat{h}$ and an adversary with realizable constraint, $A_{\mathcal{C}}$, the number of mistakes is $M(\hat{h}, A_{\mathcal{C}}, T) = \sum_{t=1}^{T} \mathbb{I}\left[Y_t \neq \hat{h}_{t-1}(X_t)\right]$.*

We say that a concept class $\mathcal{C}$ is universally online learnable in the realizable setting if and only if there is an online learning algorithm $\hat{h}$, such that $M(\hat{h}, A_{\mathcal{C}}, \infty) = \sum_{t=1}^{\infty} \mathbb{I}\left[Y_t \neq \hat{h}_{t-1}(X_t)\right]$ is finite against any realizable adversary $A_{\mathcal{C}}$.

In the agnostic setting, the adversary can arbitrarily choose the true label $Y_t$ and does not need to consider the realizable constraint. Thus, the number of mistakes made by the learning algorithm may be arbitrarily large, and we use regret to characterize the learnability for the agnostic case. Regret is usually defined as the differences between the number of mistakes made by the algorithm and the best realizable sequence, $Y^*$. In order to keep the adversary adaptive in the universal setting, we allow the adversary to choose the realizable sequence we compared with adaptively, but we need the constraint that this sequence is realizable at all times. Thus, at the beginning of each round $t$, the agnostic adversary chooses the triple $(X_t, Y_t, Y_t^*)$ adaptively and reveals $X_t$ to the learner. The learner then makes the prediction $\hat{Y}_t$ based on the entire history. After that, the adversary reveals the true label $Y_t$ to the learner for future prediction. The $Y_t^*$ is never revealed to the learner, but it should satisfy the realizable constraint, i.e., for any $t \in \mathbb{N}$, there is a concept $c \in \mathcal{C}$ such that for every $t' < t$, $Y_{t'}^* = c(X_{t'})$. We can now define regret for the agnostic setting.

**Definition 6 (Regret)** *For an online learning rule $\hat{h}$ and any adversary $A_{\mathcal{C}}$, the regret in the agnostic case is*

$$regret(\hat{h}, A_{\mathcal{C}}) = \limsup_{T \to \infty} \frac{1}{T} \mathbb{E}\left[\left(\sum_{t=1}^{T} \mathbb{I}\left[Y_t \neq \hat{h}_{t-1}(X_t)\right] - \sum_{t=1}^{T} \mathbb{I}\left[Y_t \neq Y_t^*\right]\right)\right].$$

*We say that a concept class $\mathcal{C}$ is universally online learnable in the agnostic setting if and only if there is an online learning algorithm $\hat{h}$, such that $regret(\hat{h}, A_{\mathcal{C}}) \leq 0$ against any adversary $A_{\mathcal{C}}$.*

**Bandit Feedback.** Multiclass online classification with *bandit feedback* is also a sequential game. Two players, a learner $L$ and an adversary $A$, are the same players as the players in the online learning game for the full supervision setting. The only difference between these two settings is the feedback given to the learner. In the bandit feedback setting, at round $t$, the adversary will give the learner $Y_t^{\text{bf}} = \mathbb{I}\left[\hat{Y}_t \neq Y_t\right]$ as the feedback. The history which the learner can use to inform the prediction is $(X_{<t}, Y_{<t}^{\text{bf}}) = \{(X_i, Y_i^{\text{bf}})\}_{i<t}$ instead of $(X_{<t}, Y_{<t})$. Thus, we can define the number of mistakes and the regret in the same way as the full supervision setting, but now with $\hat{h}_{t-1}(X_t) = f_t(X_{<t}, Y_{<t}^{\text{bf}}, X_t)$. We say that a concept class $\mathcal{C}$ is universally online learnable under bandit feedback in the realizable setting if and only if there is an online learning algorithm $\hat{h}$, such that $\text{M}(\hat{h}, A_{\mathcal{C}}, \infty)$ is finite against any realizable adversary $A_{\mathcal{C}}$. And we say that a concept class $\mathcal{C}$ is universally online learnable under bandit feedback in the agnostic setting if and only if there is an online learning algorithm $\hat{h}$, such that $\text{regret}(\hat{h}, A_{\mathcal{C}}) \leq 0$ against any adversary $A_{\mathcal{C}}$.

Next, we define the Littlestone tree for multiclass learning, which is a combinatorial structure used to decide whether a concept class $\mathcal{C}$ is universally online learnable with bandit feedback.

**Definition 7 (Multiclass Littlestone Tree)** A Multiclass Littlestone Tree *for a concept class $\mathcal{C}$ is a complete binary tree with the depth $d \leq \infty$. The internal nodes of that tree are labeled by elements of $\mathcal{X}$ and the edges connecting a node and its two children are labeled by two different labels from $\mathcal{Y}$, such that each finite path emanating from the root is consistent with a concept $c \in \mathcal{C}$. (Meaning that for each non-leaf node on the path, the concept $c$ labels the corresponding element of $\mathcal{X}$ with the label of the edge the path follows from that node.)*

## 3. Realizable Setting

In this section, we prove the result for the realizable setting. For brevity, we may omit "the realizable setting" in this section, when there is no ambiguity. Formally, we have the following theorem.

**Theorem 8** *For any concept class $\mathcal{C}$, the following three statements are equivalent:*

1. *$\mathcal{C}$ does not have an infinite multiclass Littlestone tree.*

2. *$\mathcal{C}$ is universally online learnable.*

3. *$\mathcal{C}$ is universally online learnable with bandit feedback.*

In order to prove this theorem, we need the following theorem from Hanneke et al. (2023b) to show the equivalence between the first and second statements.

**Theorem 9 (Hanneke et al. (2023b) Theorem 18)** *For any concept classes $\mathcal{C}$, it is universally online learnable if and only if it does not have an infinite multiclass Littlestone tree.*

**Proof of Theorem 8**

Theorem 9 shows the equivalence between the first and second statements; then, we only need to show the equivalence between the second and third statements.

To do this, we use the following way to extend the online learning algorithm with full supervision to an online learning algorithm with bandit feedback. For each $J \subseteq \mathcal{Y} \times \mathbb{N}$, such that $|J|$ is finite, we can define an expert $e^J$ based on that set $J$ as follows. $Y_t^{e^J} = e^J(X_{<t}, Y_{<t}^{e^J}, X_t)$. Here $e^J(X_{<t}, Y_{<t}^{e^J}, X_t) = f_t^{\mathrm{fs}}(X_{<t}, Y_{<t}^{e^J}, X_t)$ if $(\tilde{Y}_t, t) \notin J$ for any $\tilde{Y}_t \in \mathcal{Y}$, otherwise $Y_t^{e^J} = \tilde{Y}_t$. Then according to the theorem 9, for any realizable sequence, there exists a set $J$, such that $f^{\mathrm{fs}}$ only makes mistakes at round $t$, when $(\tilde{Y}_t, t) \in J$. In the meanwhile, $Y_t = \tilde{Y}_t$ if and only if $(\tilde{Y}_t, t) \in J$. Thus, for every realizable sequence, there is an expert $e^J$ such that $Y_t^{e^J} = Y_t$ for all $t$. Notice that we only have countably infinite different $J$'s. Thus, we can index those experts by natural numbers and we follow the prediction of expert $i$, until it makes a mistake and switch to following the prediction of expert $i + 1$. After a finite number of mistakes, we will switch to the expert whose prediction is the same as the true label and it will never make mistakes anymore. Therefore, we can extend the algorithm with full supervision to an algorithm with bandit feedback.

Then notice that if we have an online algorithm $\hat{h}^{\mathrm{bf}}$ for the bandit feedback setting, we can use it as a subroutine and change the true label to the bandit feedback to obtain an online learning algorithm for full supervision and keep the number of mistakes unchanging. Therefore, the statement that we want to prove holds. ∎

After that we can provide the formal proof for our main theorem for the realizable case, Theorem 1.

**Proof of Theorem 1** According to Theorem 8, the first bullet of Theorem 1 is valid. To prove the second bullet, we slightly extend the proof in the work of Hanneke et al. (2023b) to handle randomized learners. We build an adversary that forces any learner to make a linear expected number of mistakes. Consider the infinite multiclass Littlestone tree, the adversary takes a random walk from the root of that tree to generate the realizable sequence $(X_{<\infty}, Y_{<\infty})$. Then, in each round $t$, no matter what prediction the learner makes, it will make a mistake with probability $\frac{1}{2}$. By the linearity of the expectation, this adversary forces any learner to make a linear expected number of mistakes. That finishes the proof.

∎

## 3.1. Alternative Proof of Theorem 8

We can also extend any algorithm for full supervision to an algorithm for bandit feedback and preserve that the number of mistakes of the algorithm is finite by using the technique used in the work of Long (2017) and it may be of independent interest.

---

**Universal Multiclass Learning Algorithm with Bandit Feedback**

1. Parameter: $\alpha$, $q_y = \frac{1}{i(i+1)}$.

2. Initialize $E_0 \leftarrow \{(\hat{h}_0^{\text{fs},(1)}, w_0^{(1)} = 1)\}$.

3. For $t = 1, 2, 3, \ldots$:

4.     Predict $\hat{Y}_t = \text{argmax}_y \sum_i w_{t-1}^{(i)} \mathbb{I}\left[\hat{h}_{t-1}^{\text{fs},(i)}(X_t) = y\right]$.

5.     If $Y_t = \hat{Y}_t$:

6.         For every $i$, such that $(\hat{h}_{t-1}^{\text{fs},(i)}, w_{t-1}^{(i)}) \in E_{t-1}$:

7.             Update $\hat{h}_{t-1}^{\text{fs},(i)}$ to $\hat{h}_t^{\text{fs},(i)}$ with $(X_t, Y_t)$ and put $\{(\hat{h}_t^{\text{fs},(i)}, w_t^{(i)})\}$ into $E_t$.

8.     Else $Y_t \neq \hat{Y}_t$:

9.         For every $i$, such that $(\hat{h}_{t-1}^{\text{fs},(i)}, w_{t-1}^{(i)}) \in E_{t-1}$:

10.         If $\hat{h}_{t-1}^{\text{fs},(i)}(X_t) \neq \hat{Y}_t$

11.             Let $\hat{h}_t^{\text{fs},(i)} \leftarrow \hat{h}_{t-1}^{\text{fs},(i)}$ and put $(\hat{h}_t^{\text{fs},(i)}, w_t^{(i)} = w_{t-1}^{(i)})$ into $E_t$.

12.         Else, $\hat{h}_{t-1}^{\text{fs},(i)}(X_t) = \hat{Y}_t$,

13.             For $y \in \mathcal{Y}$, such that $y \neq \hat{Y}_t$,

14.                 Create a copy of $\hat{h}_{t-1}^{\text{fs},(i)}$ and update it to $\hat{h}_t^{\text{fs},(i')}$ with $(X_t, y)$ and put $\{(\hat{h}_t^{\text{fs},(i')}, w_t^{(i')} = \alpha q_y w_{t-1}^{(i)})\}$ into $E_t$.

---

**Proof of Theorem 8 by using the method from Long (2017)** To prove this, we first show how to use algorithm $\hat{h}^{\text{fs}}$ as a subroutine to build algorithm $\hat{h}^{\text{bf}}$. The learning algorithm $\hat{h}^{\text{bf}}$ keeps a list of copies of $\hat{h}^{\text{fs}}$ that have different inputs, and all of these copies are treated as experts. Each expert is assigned a weight. In each round, $\hat{h}^{\text{bf}}$ makes its prediction based on the weighted majority of the opinions of experts. Every time $\hat{h}^{\text{bf}}$ makes a mistake, for each expert that agrees with its prediction, the weight of that expert is multiplied by a penalty. The detailed algorithm is the following 3.1.

An important observation is that for every realizable sequence generated by the adversary $A_{\mathcal{C}}$, there exists a copy of $\hat{h}^{\text{fs}}$ that always takes the correct information, marked as $\hat{h}^{\text{fs},*}$. Because we can take the online learning algorithm for full supervision $\hat{h}^{\text{fs}}$ which only makes finite mistakes for any adversary with full supervision, $\hat{h}^{\text{fs},*}$ makes only finite mistakes, marked as $m$. In addition, we can note the correct labels for these rounds as $\{Y_{t_1}, \cdots, Y_{t_m}\}$. The weight of that copy is at least $\alpha^m \prod_{j=1}^m q_{Y_{t_j}}$. Thus, there exists a constant $\varepsilon > 0$ such that the weight of $\hat{h}^{\text{fs},*}$ is greater than or

equal to $\varepsilon$ in any round. (We can take $\varepsilon = \alpha^m \prod_{j=1}^m q_{Y_{t_j}}$ as an example.) Therefore, the total weight of all copies in round $n$, $W_n$, is greater than or equal to $\varepsilon$ for all $n$.

We can look at the total weight in another way and build an upper bound for that based on the number of mistakes $\hat{h}^{\mathrm{bf}}$ make. According to the algorithm we described above, if $\hat{h}_{t-1}^{\mathrm{bf}}(X_t) \neq Y_t$, we have the following inequality between $W_t$, the total weight in round $t$, and $W_{t-1}$, the total weight in round $t-1$:

$$W_t \leq W_{t-1} - W_{t-1}^{(\hat{Y}_t)} + \alpha(1 - q_{\hat{Y}_t})W_{t-1}^{(\hat{Y}_t)} \tag{1}$$

Here $W_{t-1}^{(\hat{Y}_t)}$ is the sum of the weights of the copies whose prediction is $\hat{Y}_t$ in round $t$. This inequality comes from the step where we create the copies. In that step, for each copy whose prediction is $\hat{Y}_t$, and for each other label $y$, we create a new copy and multiply its weight by $\alpha q_y$. Thus, the total weight of the copies with the prediction $\hat{Y}_t$ is multiplied by $\alpha(\sum_{y \neq \hat{Y}_t} q_y)$, which is $\alpha(1 - q_{\hat{Y}_t})$. From the previous paragraph, we know that there is a copy whose weight is always greater than or equal to $\varepsilon$. Due to the algorithm, $\hat{Y}_t$ is the label with the highest weight in round $t$. Thus, $W_{t-1}^{(\hat{Y}_t)} > \varepsilon$ as well. So, we have:

$$W_t \leq W_{t-1} - (1 - \alpha)W_{t-1}^{(\hat{Y}_t)} - \alpha q_{\hat{Y}_t} W_{t-1}^{(\hat{Y}_t)} \leq W_{t-1} - (1 - \alpha)\varepsilon - \alpha q_{\hat{Y}_t}\varepsilon \tag{2}$$

So after $t$ mistakes, we will have the total weight $W_t \leq W_0 - t(1 - \alpha)\varepsilon - \alpha(\sum_{j=1}^t q_{\hat{Y}_{t_j}})\varepsilon$. For the sake of contradiction, suppose that $\hat{h}^{\mathrm{bf}}$ makes infinite mistakes, then we can take $t > \frac{1}{(1-\alpha)\varepsilon}$. Notice that $W_0 = 1$, so we have $W_t < 0$, which contradicts the fact that $W_n \geq \varepsilon$ for all $n$. Thus, $\hat{h}^{\mathrm{bf}}$ only makes finite mistakes against that adversary. This proof holds for every adversary. ∎

### 3.2. Universal Bandit Littlestone Tree

In this subsection, we introduce an extension of the bandit Littlestone tree appropriate for the universal setting, called the universal bandit Littlestone tree. Then, we show this combinatorial complexity structure characterizes universal online learnability with bandit feedback even when the label space is infinite (possibly even uncountable).

**Definition 10 (UBL Tree)** A universal bandit Littlestone tree *(UBL-tree for brevity) for a concept class $\mathcal{H}$ is a perfect tree $T$, with depth $d \leq \infty$. The internal nodes of that tree are labeled with elements of $\mathcal{X}$ and the edges connecting a node and its children are labeled with elements of $\mathcal{Y}$, without repetition. By saying a tree is "perfect", we mean that for every element $y \in \mathcal{Y}$, there exists an edge labeled $y$ from every internal node. We say that a BL tree $T$ is shattered by a concept class $\mathcal{H}$, if for every path $P$ starting from the root, there exists a sequence of concepts $h_i \in \mathcal{H}$ for all $i \in \mathbb{N}$, such that $h_i(x_{v_j})$ disagrees with the edge in $P$ that connects $v_j$ and its child for all $j \leq i$, and $h_{i+1}$ agrees with $h_i$ on $x_{v_j}$ for all $j \leq i$.*

We have the following theorem:

**Theorem 11** *The following two statements are equivalent:*

1. *$\mathcal{H}$ does not have an infinite UBL tree.*

2. *$\mathcal{H}$ is universal online learnable in the realizable setting.*

In order to prove the theorem, we need the following lemma:

**Lemma 12** *If a concept class $\mathcal{H}$ has an infinite Littlestone tree, it also has an infinite UBL tree.*

**Proof** Take the infinite Littlestone tree $T$, we can transform it into an infinite UBL tree $T'$ by the following procedure: starting from the root, for each node $v \in T$, find the nodes $v' \in T'$ labeled by $x_v$ and create many edges, such that every $y \in \mathcal{Y}$ has an edge labeled by it without repetition. If $v$ has two children $v_{y_1}$ and $v_{y_2}$, which are labeled by $x_{v_{y_1}}$ and $x_{v_{y_2}}$, then for $T'$, the child adjacent with the edge $y_1$ is labeled by $x_{v_{y_2}}$; the child adjacent with the edge $y_2$ is labeled by $x_{v_{y_1}}$ and the rest children can pick a label from $\{x_{v_{y_1}}, x_{v_{y_2}}\}$. Thus, we can create an infinite UBL tree from an infinite Littlestone tree, which finishes the proof. ∎

**Proof of Theorem 11** Consider the following UBL game: There are two players, $P_A$ and $P_B$. On each round $t$, player $P_A$ proposes an $x_t \in \mathcal{X}$, then player $P_B$ proposes a $y_t \in \mathcal{Y}$; player $P_A$ wins if and only if there is a labeling $y_t^*$ of the infinite sequence of $x_t$'s which avoids all the $y_t$ labels ($y_t^* \neq y_t$), and is realizable by the concept class $\mathcal{H}$ (in the usual sense that all finite prefixes are realizable).

Notice that $P_A$ has a winning strategy if and only if there is an infinite UBL tree. If $P_B$ has a winning strategy $g_U$, we can run the following algorithm to guarantee finite mistakes.

---

**Universal Multiclass Learning Algorithm with Bandit Feedback from Winning Strategy $g$**

1. Initialize $U = \emptyset$.

2. For $t = 1, 2, 3, \ldots$:

3.     Predict $\hat{Y}_t = g_U(X_t)$.

4.     If $Y_t \neq \hat{Y}_t$:

5.         Let $U = U \cup \{(X_t, \hat{Y}_t)\}$ and update the game.

---

Then we need to show the UBL game is determined (i.e., if there is no infinite UBL tree, then $P_B$ has a winning strategy, so that in any case one of the two players must have a winning strategy). For countable label spaces, because of Lemma 12, if it does not have an infinite UBL tree, we know

it does not have an infinite Littlestone tree and refer to the proof of Theorem 8, we have countable experts $E = e_1, e_2, \cdots$ which enumerate all possible realizable sequences for the data sequence $\{X_i\}_{i \in \mathbb{N}}$. Thus, for this case, if $P_A$ does not have a winning strategy, $P_B$ has a winning strategy, which is to pick $y_t = e_t(X_t)$. For uncountable label spaces[4], we always have an infinite UBL tree, where every node is labeled by the same $x$ (chosen to be such that its set of realizable labels is uncountable), because for every infinite branch, we can always find a label $y$ which does not appear in the branch and there exists $h \in \mathcal{H}$ with $h(x) = y$. Hence, $P_A$ always has a winning strategy. Therefore, the UBL game is determined in both cases, and that finishes the proof. ∎

This theorem provides a more thorough characterization for universal multiclass online learnability with bandit feedback in the realizable setting when the label spaces can even be uncountable.

## 4. Agnostic Setting

In this section, we discuss the results of the agnostic case. We can extend the algorithm for the realizable setting to the algorithm for the agnostic setting and guarantee that the regret of the algorithm is low. To reach this target, we need to build a set of experts such that there is an expert whose regret is finite against any adversary. We also need an online learning algorithm with experts' advice using bandit feedback such that the difference between the number of mistakes made by the algorithm and the best expert grows sublinearly.

Then we can first define the set of experts by using the following recursive function. For each set $J \subset \mathbb{N}$, satisfies that $|J|$ is finite, we define $e_t^J = e^J(X_{<t}, \tilde{Y}_{<t}^{\mathrm{bf}}, X_t) = f_t^{\mathrm{bf}}(X_{<t}, \tilde{Y}_{<t}^{\mathrm{bf}}, X_t)$, where $\tilde{Y}_t^{\mathrm{bf}} = 1$ if $t \in J$ and 0 otherwise.

**Lemma 13** *For any agnostic adversary $\mathcal{A}_C$ which generates $(X_{<\infty}, Y_{<\infty}, Y_{<\infty}^*)$, there is a set $J$, such that $e_t^J \neq Y_t^*$ if and only if $t \in J$.*

**Proof** Due to Theorem 1, if a concept class $\mathcal{C}$ does not have an infinite multiclass Littlestone tree, we have an online learning algorithm with bandit feedback $\hat{h}^{\mathrm{bf}}$ against a realizable adversary, $A_C^R$, which generates $(X_{<\infty}, Y_{<\infty}^*)$ and makes only finite mistakes. Thus, we can let set $J$ be the set of all indexes of the round that $\hat{h}^{\mathrm{bf}}$ makes a mistake. In other words, $j \in J$ if and only if $f_t^{\mathrm{bf}}(X_{<t}, Y_{<t}^{\mathrm{bf}}, X_t) \neq Y_t^*$. Here $Y_t^{\mathrm{bf}} = \mathbb{I}\left[f_t^{\mathrm{bf}}(X_{<t}, Y_{<t}^{\mathrm{bf}}, X_t) \neq Y_t^*\right]$. Notice that the expert $e^J$ defined above simulates $\hat{h}^{\mathrm{bf}}$ running against the realizable adversary $A_C^R$. Thus, $e_t^J \neq Y_t^*$ if and only if $t \in J$. That finishes the proof. ∎

After defining the experts, we need the following result from the learning from experts' advice problem. The algorithm EXPINF from the work of Blanchard, Hanneke, and Jaillet (2022) has the following property:

**Lemma 14 (Blanchard et al. (2022), Corollary 3.5)** *There is an online learning rule EXPINF using bandit feedback such that for any countably infinite set of experts $\{e^{(1)}, e^{(2)}, \dots\}$ (possibly ran-*

---

4. $\sup_x |\{h(x) : h \in \mathcal{H}\}|$ is uncountable, otherwise, we can reduce it to the countable label spaces case.

*domized), with probability one on the learning and the experts, there exists $\hat{T}$ such that for any $T \geq 1$,*

$$\max_{1 \leq i \leq T^{\frac{1}{8}}} \sum_{t=1}^{T} \mathbb{I}\left[\textit{EXPINF}(X_t) \neq Y_t\right] - \mathbb{I}\left[e_t^{(i)} \neq Y_t\right] \leq \hat{T} + cT^{\frac{3}{4}}\sqrt{\ln T}\ln T.$$

*Here $c > 0$ is a universal constant.*

This lemma shows that we have an online learning algorithm that accepts bandit feedback and a countable infinite set of experts and makes sure the difference between the number of mistakes made by the algorithm and the best expert is sublinear.

Combining all the facts above, we can prove the following main theorem:

**Theorem 15** *If a concept class $\mathcal{C}$ does not have an infinite multiclass Littlestone tree, there exists a learning algorithm $\hat{h}^{bf}$ whose regret is less than or equal to $0$ almost surely against any adversary $A_\mathcal{C}$.*

**Proof** For any adversary $A_\mathcal{C}$ and any multiclass online learning algorithm with bandit feedback $\hat{h}^{\text{bf}}$, we have

$$\limsup_{T \to \infty} \frac{1}{T} \sum_{t=1}^{T} \left( \mathbb{I}\left[\hat{h}^{\text{bf}}_{t-1}(X_t) \neq Y_t\right] - \mathbb{I}\left[Y_t^* \neq Y_t\right] \right)$$

$$= \limsup_{T \to \infty} \frac{1}{T} \sum_{t=1}^{T} \left( \mathbb{I}\left[\hat{h}^{\text{bf}}_{t-1}(X_t) \neq Y_t\right] - \mathbb{I}\left[e_t^J \neq Y_t\right] + \mathbb{I}\left[e_t^J \neq Y_t\right] - \mathbb{I}\left[Y_t^* \neq Y_t\right] \right).$$

Then according to Lemma 13, for the adversary $A_\mathcal{C}$, there is an expert $e^J$ such that $e_t^J \neq Y_t^*$ if and only if $t \in J$. We also know that the set of all finite subsets of $\mathbb{N}$ is countably infinite. Then according to the Lemma 14, by taking the EXPINF with the experts defined by Lemma 13 as the online learning algorithm $\hat{h}^{\text{bf}}$, we have:

$$RHS = \limsup_{T \to \infty} \frac{1}{T} \left( \sum_{t=1}^{T} \left( \mathbb{I}\left[\hat{h}^{\text{bf}}_{t-1}(X_t) \neq Y_t\right] - \mathbb{I}\left[e_t^J \neq Y_t\right] \right) + \sum_{t=1}^{T} \left( \mathbb{I}\left[e_t^J \neq Y_t\right] - \mathbb{I}\left[Y_t^* \neq Y_t\right] \right) \right)$$

$$\leq \limsup_{T \to \infty} \frac{1}{T} \left( \hat{T} + cT^{\frac{3}{4}}\sqrt{\ln T}\ln T + \sum_{t=1}^{T} \left( \mathbb{I}\left[e_t^J \neq Y_t\right] - \mathbb{I}\left[Y_t^* \neq Y_t\right] \right) \right).$$

Due to Lemma 13, $e_t^J \neq Y_t^*$ if and only if $t \in J$. Thus, we have $\sum_{t=1}^{T} \left( \mathbb{I}\left[e_t^J \neq Y_t\right] - \mathbb{I}\left[Y_t^* \neq Y_t\right] \right) \leq |J|$ and then

$$RHS \leq \limsup_{T \to \infty} \frac{1}{T} \left( \hat{T} + cT^{\frac{3}{4}}\sqrt{\ln T}\ln T + |J| \right) = 0.$$

This is because $\hat{T}$ and $|J|$ are both constant, we can take $T$ be large enough, such that $T^{\frac{3}{4}}\sqrt{\ln T}\ln T$ dominates and also $T^{\frac{1}{8}}$ is greater than the index of the expert $e^J$. Thus, the limitation is 0.

Then by Fatou's lemma,

$$\limsup_{T\to\infty} \frac{1}{T}\mathbb{E}\left[\left(\sum_{t=1}^{T}\mathbb{I}\left[Y_t \neq \hat{h}_{t-1}^{\text{bf}}(X_t)\right] - \sum_{t=1}^{T}\mathbb{I}\left[Y_t \neq Y_t^*\right]\right)\right]$$

$$\leq \mathbb{E}\left[\limsup_{T\to\infty} \frac{1}{T}\left(\sum_{t=1}^{T}\mathbb{I}\left[Y_t \neq \hat{h}_{t-1}^{\text{bf}}(X_t)\right] - \sum_{t=1}^{T}\mathbb{I}\left[Y_t \neq Y_t^*\right]\right)\right]$$

$$\leq 0.$$

∎

**Proof of Theorem 3** Theorem 15 shows the first bullet of Theorem 3. For the second bullet, the realizable setting is a special case of the agnostic setting. Hence, if we have an online learning algorithm that takes bandit feedback and its regret is smaller than or equal to 0 against all adversaries, we can take the agnostic adversary such that $Y_t = Y_t^*$ for all $t$, this algorithm will be an online learning algorithm only makes sublinear mistakes against any realizable adversaries. Thus, according to Theorem 1, if a concept class $\mathcal{C}$ has an infinite multiclass Littlestone tree, there is an adversary forces any online learning algorithm makes linear expected regret. Otherwise, there is an online learning algorithm for the realizable case which only makes sublinear expected number of mistakes, which contradicts Theorem 1. ∎

## 5. Conclusion, Discussion, and Future Directions

In this paper, we study multiclass online learning under bandit feedback within the universal learning framework. Our findings reveal an unexpected exact equivalence between multiclass online learning with bandit feedback and full supervision within this framework, both in the realizable and agnostic settings, even when the effective label space is countably infinite. In particular, we establish that in both settings, a concept class $\mathcal{C}$ is learnable if and only if it does not have an infinite multiclass Littlestone tree.

At this point, we highlight some recommended directions and suggestions for future investigations.

- Firstly, we think it is interesting to study the framework of Hanneke, Moran, Raman, Subedi, and Tewari (2023a) under bandit feedback. Moreover, we conjuncture that in the multiclass learning with i.i.d instances within the universal learning framework, bandit feedback and full supervision are also exactly equivalent, even when the effective label space is countably infinite.

- Secondly, it would be of significant interest to explore alternative forms of feedback beyond bandit feedback in the multiclass online learning within the universal learning framework.

These feedbacks includes, apple tasting Helmbold, Littlestone, and Long (2000), dynamic pricing Cesa-Bianchi and Lugosi (2006), police and criminals Alon, Cesa-Bianchi, Dekel, and Koren (2015), matching pennies Lattimore and Szepesvári (2020), and feedback graphs Mannor and Shamir (2011). More specifically, for which types of feedback do we have a similar equivalence as for bandit feedback?

## Acknowledgments

We would like to thank an anonymous reviewer for his/her insightful suggestion on the proof of Theorem 8 and the definition of UBL tree Definition 10.

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
