# OpenReview forum: "For Universal Multiclass Online Learning, Bandit Feedback and Full Supervision are Equivalent"
_algorithmiclearningtheory.org/ALT/2025/Conference — ALT 2025_

### Official Review · Reviewer_1b6p · 2024-11-05
**A good start but I would like to see more**

**Rating:** 4
**Confidence:** 4

**Review:**

The paper studies a variant of multiclass online learning with two properties: a) only bandit feedback is available to the learner b) the learner is supposed to make only finite number of mistakes but without requiring an uniform bound on mistakes. The authors call this universal learning, after Bousquet et al 2021, although it would be perhaps more appropriate to just call it the non-uniform setting, per analogy to non-uniform PAC learning. There are two related contributions here: a) they show that in non-uniform multiclass online learning, learnability with full feedback already implies learnability with bandit feedback in the realizable case and b) the same holds for agnostic learning, in the sense of sublinear regret. This in turn allows the authors to present new equivalences between combinatorial characterization of a class (ordinal Littlestone dimension) and nonuniform bandit learnability.

The paper is clearly relevant to ALT community, while the results and contributions are stated clearly. I have some reservations about the significance or novelty of the contribution. While the results seems to be new (in the sense that they were not claimed by anyone before), they are not very surprising and the proofs are simple applications of some existing techniques with very little of new mathematical insights. Especially the part about realizable case seems rather obvious and it feels to me like something on the level of textbook exercise rather than a main contribution of a research paper. If H has does not shatter an infinite Littlestone tree, then the learner has effectively only countably many hypotheses to consider. Then the simplest nonuniform learning algorithm can be used: enumerate this countable set of possible hypotheses and try them one by one. Obviously, for this to work we do not need the full feedback, the information that a given hypothesis is wrong is enough to reject it. This kind of ideas has been around for a long time, probably already in the work of Gold and Putnam.

On the other hand, the presentation of the proofs in the agnostic part could be improved, given that the main idea there is also simple, if Lemma 11 is taken as black-box. I believe that the result claimed for agnostic setting is correct and I see how to prove it but this is mostly because I can reconstruct the idea, somehow despite the way its written. To summarize, the main weaknesses I see are: a) the contribution is small, the results are quite straightforward b) the presentation of some proofs could be improved. At the same time, I would like to encourage the authors to continue working on this setting. In my opinion, the current submission forms a nice start for something that could be expanded to a very good paper.

Some minor comments:
Page 12, Section 4, second par. 'For each J \subset \mathbb{N}, satisfied that |J| is finite' -> For each finite J \subset \mathbb{N}
further in that sentence: what is f^{bf} here? I assume that the authors explain how to define experts starting from some arbitrary f^{bf} but maybe there is something more going on here

Page 12, proof of Lemma 10. How are f and h related? Are they the same learner?

**Paper Award:**

No

---

### Official Review · Reviewer_xq6u · 2024-11-08

**Rating:** 7
**Confidence:** 3

**Review:**

Summary

This paper considers the universal multiclass online learning under bandit feedback setting. This is a sequential setting where in each round, a learner receives an instance and subsequently chooses a label from a potentially unbounded number of labels, and subsequently the learner sees whether or not the chosen label was the correct label. There are two versions of this setting: the realisable setting, where there is a concept from a concept class that perfectly predicts labels given the instances, and the agnostic setting, where such a concept does not exist. The goal in the realisable setting is to design learning algorithms that make a finite number of mistakes as the number of rounds goes to infinity. The goal in the agnostic setting is to have small regret relative to the best concept in the class. The authors show that if the concept class has a finite littlestone tree, then both in the realisable and agnostic settings then the concept class is learnable.

Strengths and weaknesses.

Overly simplified, in the realisable setting, the question is as follows: given enough interaction with the environment, can we identify the concept that generates the true labels? Surprisingly, this work answers this question by showing that if the concept class has a finite littlestone tree, that this is indeed the case. The strategy to show that this is the case is remarkably straightforward, which I consider a plus.

As written by the authors, this shows a stark contrast between multiclass online learning in the uniform framework and in the universal learning framework: in the uniform framework the concept class that is learnable even in the full information setting is considerably smaller. Even though the question that is asked in the uniform framework is considerably harder by definition I think that the difference as shown in this paper is very interesting. Specifically, for me this is the clearest demonstration that I have read about the differences between the uniform and the universal framework.

The main downside about this work is how the paper is structured. Even though the results are introduced early on in the paper, the definition and goal of the setting are only given on page 9. This for me makes the paper unnecessarily hard to read, as I had to first wrestle through the introduction without really knowing what the setting was. Instead, if the setting was introduced early on in the paper, it would have been easier for me to parse the paper. It would have also been nice to state the goal in the uniform framework more clearly, as this would have made it easier for me to understand why there is such a big difference in what is learnable for the two settings.

**Paper Award:**

No

---

> ### Author Response · Authors · 2024-11-25
>
> We are delighted that the reviewer found our work surprising and our results very interesting, and found that this is the clearest demonstration that the reviewer has read about the differences between the uniform and the universal framework. Below, we address one issue that the reviewer raised.
>
> - First of all, we especially thank the reviewer for sharing thoughts on improving the presentation of our paper: We will make the following changes in the camera-ready version. In the introduction, we will make sure to state our setting clearly, along with its goal. In addition, we will add a subsection in the introduction to compare uniform, non-uniform, and universal settings.

---

### Official Review · Reviewer_TobW · 2024-11-08
**Bandit multiclass universal learnability**

**Rating:** 9
**Confidence:** 4

**Review:**

The paper studies multiclass classification under bandit feedback in
the setting of universal learning, a recently proposed learning
framework that is in stark distinction with the more classical uniform
learning. The paper shows an interesting and perhaps surprising
result, that in the framework of universal learning, there is no
distinction between learnability under bandit feedback and under full
information. In particular, universal learning is possible whenever
the concept class does not admit a littlestone tree of infinite
depth. The surprising part is that this is different from what happens
under uniform learning: for uniform learning, learning is not possible
under bandit feedback if the "effective label space" is (countably)
infinite.

The techniques incorporate some newer ideas, like the recent works on
universal learning, with some older ideas like multiplicative weight
updates and EXP4 (with some clever modifications). I looked at the
proofs briefly and they seem correct.

Overall, I have no major issues with the paper and think it is clearly
above the bar for the conference. The topic and results are timely and
certainly of interest to the community, and the techniques seem
non-trivial. I only have some minor comments:

- Can get the motivation for defining the triplet sequences
  (X_t,Y_t,Y_t^\star) but what goes wrong at the technical level if
  one does not do this? Is this an issue if the adversary is oblivious
  or is this just an issue of adaptivity?

- For the realizable setting is it possible to derive a quantitative
  mistake bound for the bandit setting (say as a function of the
  mistake bound for the full information setting)? Maybe doing this in
  the special case of a finite label space and taking q_Y = 1/|Y|
  would be illuminating? It seems like it is possible in this case?

**Paper Award:**

No

---

> ### Author Response · Authors · 2024-11-25
>
> We are delighted that the reviewer found our topic and results timely and certainly of interest to the community, and found that our work is clearly above the bar for the conference. Below, we address two minor comments that the reviewer raised.
>
> - The formulation of the appropriate "comparator" in the definition of agnostic universal online learning is a subtle issue, representing one of the key contributions of this work.  This is not so much an issue of oblivious vs adaptive.  Rather, it is about expressing a notion of "comparator" in the definition of "regret" to match the universal quantification "forall sequences".  Consider what would happen if we used the traditional "best in class" comparator: i.e., $min_{c \in H} \sum_{t=1}^T \mathbb{I}[c(X_t) \neq Y_t]$.  Learnability (even with full supervision) would then still be characterized by having finite Littlestone dimension.  To see this, if $LD(H)=\infty$, consider a sequence of shattered Littlestone trees of depths $k^{2k}$, $k=2,3...$, and set the sequence $(X_t,Y_t)$ so the first $2^4$ examples follow a random branch in the tree of depth $2^4$, then the next $3^6$ examples follow a random branch in the tree of depth $3^6$, and so on.  For each $T = \sum_{j=2}^k j^{2j}$, the learner has erred on at least $(1/2)k^{2k} \geq (1/2)T-o(T)$ examples, yet there is a concept correct on the last $k^{2k}$ examples which therefore errs on at most $\sum_{j \leq k-1} j^{2j} < (1/k)T$, so the regret fails to be sublinear in $T$.  A naive way to formulate the universal objective would be "for all concepts c in H, the algorithm makes sublinearly more mistakes than c as $T \to \infty$".  However, restricting $c$ to $H$ is too strong, as we wish to only restrict the comparator labels to be realizable on all finite prefixes (not the infinite sequence) matching the notion of realizability studied in the realizable case.  By allowing the adversary to generate an arbitrary realizable sequence $Y^*_t$ (hidden from the learner) as the comparator, we elegantly allow for this relaxed constraint on the comparator.
>
> - Yes, if we consider the setting with finite label space, then we can prove that for any adversarial sequence S, we do not make more than $k \times \log(k) \times M(S)$ number of mistakes, where k is the number of labels and $M(S)$ is the number of mistakes that the full supervision algorithm makes on S. We will mention this statement as a remark in the camera-ready version.

---

### Author Rebuttal · Authors · 2024-11-21

We thank the reviewers for their efforts, and the kind and helpful comments and feedback.

We have responded to each review individually in our posted official comments.

---

> ### Author Rebuttal · Authors · 2024-11-21
>
> For reviewer #3. We appreciate your helpful comments. But we would like to point out that it seems that you misunderstood our problem setting.
>
> First, we think this paper (https://arxiv.org/pdf/2312.00170) is the non-uniform setting you mentioned. According to that paper, there are two main differences between the non-uniform setting and our universal setting.
> 1) The non-uniform setting, even though, is not trying to build a uniform mistake bound on all concepts, still asks for a mistake bound only depending on the concept. However, in our universal setting, we only care about the number of mistakes to be finite for a specific sequence.
> 2) The definition of the realizable case in the non-uniform setting is different from our universal setting. In the non-uniform setting, they say a sequence is realizable if and only if there is a concept $h^*$, such that for all $i$, $Y_i = h^*(X_i)$, in other words, the whole sequence is realizable. However, in our setting, we only need all finite prefixes of the sequence to be realizable, we will call this sequence a realizable sequence. Thus, the learnability for the non-uniform setting is not comparable to the learnability of our setting. Even though we can change our definition of realizable sequence to the definition in the non-uniform setting, it is still unclear whether those two settings are the same, as the definitions of learnability are still different, as we have mentioned in the first part.
>
> Secondly, you mentioned, "If H has does not shatter an infinite Littlestone tree, then the learner has effectively only countably many hypotheses to consider. ". This is not true as well. For example, the concept class of singleton defined on R, in other words, the class of function $f_x$ such that for all $x' \in \mathbb{R}$, $f_x(x') = 1\text{ iff }x' = x,\text{ otherwise }0.$. This concept class has no infinite Littlestone tree, but it contains uncountably infinite hypotheses.
>
> Thus, whether the common non-uniform learning algorithm can be used in our setting is unclear. We are using the weighted majority algorithm with non-uniform initial weights to solve this problem, this tool is brand new and the result is very surprising.
>
> For your minor comments, we provide the alternative definition for the online learning rules, f and \hat{h} are the same online learning rule, we will use f if we need to state what the history of the learning algorithm is using. We expressed this in the last paragraph on page 8.

---

> > ### Comment · Reviewer_1b6p · 2024-11-21
> > **there is no misunderstanding**
> >
> > I understand the setting very well and I was not referring to the NeurIPS paper. In fact, it seems to me that the authors might be misunderstanding 'universal learning', which is about learning rates in **probabilistic setting** (and not adversarial one). The setting they discuss appears in the STOC paper (Bousquet et al) but it's not what is called 'the universal learning' in that paper.
> >
> > From the abstract of the STOC paper: 'More precisely, we consider the problem of universal learning, which aims to understand the performance of learning algorithms on every data distribution, but without requiring uniformity over the distribution. ' You define universal learning as a setting with an Adversary.
> >
> > The authors also misunderstood my other comment. I never said that having an ordinal Littlestone dimension implies countability of the whole class. I said: the learner has effectively only countably many hypotheses to consider. This is because the Adversary is giving a countable sequence of points and on this sequence only countably many functions can be realized (otherwise, we could build an infinite Littlestone tree). The only subtlety is that you need to build the enumeration in the online manner, but it's not a major issue.
> >
> > Yet another direct proof could be probably obtained by using Bandit SOA from Hanneke et al 2015 paper "Multiclass Learnability and the ERM Principle" (which the authors don't even cite!). This variant of SOA operates on a bandit analog of Littlestone trees. One needs only to observe that having an infinite Bandit Littlestone tree implies having an infinite Littlestone tree. Therefore, Bandit SOA makes a finite number of mistakes on a class with no infinite Littlestone tree.
> >
> > To summarize, I understand Theorem 8 might seem 'very surprising' for some people, but for someone who have spent some time thinking about these kind of problems, this is really neither shocking or impressive. I strongly believe that the current contribution is not enough for a good conference.

---

> > > ### Author Response · Authors · 2024-11-25
> > >
> > > Thank you for your response.  It helped clarify quite a lot about your comments, which we believe we now fully understand and appreciate.
> > >
> > > Let us start by mentioning what we find most interesting/surprising about the result, as related to the following comment:
> > >
> > > > "having an infinite Bandit Littlestone tree implies having an infinite Littlestone tree. Therefore, Bandit SOA makes a finite number of mistakes on a class with no infinite Littlestone tree."
> > >
> > > It is important to note that we allow infinite label spaces, in which case there is typically an infinite BL-tree.  Consider even the class H of all *constant* functions {$ x \mapsto c_y(x) = y : y \in Y $} with an infinite $Y$ space (note Ldim = 1): for any x, the infinite BL-tree with x in every node is shattered by H (i.e., for every finite-depth path from the root, some concept in H is incorrect on all edge labels along the path: namely, any $c_y$ with $y$ not among those edge labels).
> > >
> > > Indeed, to us, this hits on the most interesting (and surprising) point of our main results.  The infinite label setting with bandit feedback is trivially non-learnable in the uniform setting, yet yields a very reasonable theory of learnability in the universal setting (in fact, equivalent to the full-supervision setting, unlike the uniform analysis when the label space is infinite).
> > >
> > > Another interesting aspect we want to point out is that, in the agnostic setting, even the formulation of the objective is new in this work, and required consideration of subtle issues regarding what the appropriate "comparator" should be in the definition of regret, so that the "for all comparators" universal style of analysis is possible, rather than the uniform "best in class" notion of regret (as the latter is still characterized by finite Littlestone dim).  We have found an elegant way to express the appropriate objective, which itself should be of value to future work on this subject.
> > >
> > >
> > > > "universal learning"
> > >
> > > Not to hammer the terminology issue too much, but let us explain the background:
> > > From the paragraph above Theorem 3.1 page 15 of the Bousquet, Hanneke, Moran, van Handel, Yehudayoff (2021) arXiv paper (https://arxiv.org/pdf/2011.04483):
> > > "The above notion of learnability may be viewed as a universal analogue of the uniform mistake bound model of Littlestone (1988)"
> > >
> > > The "universal" terminology of Bousquet, et al. refers to the "$\forall$" universal quantifier: for all realizable distributions, the error converges at some rate (though perhaps not uniformly so over distributions); analogously, we are interested in whether, for all realizable data sequences, the number of mistakes is finite (though perhaps not uniformly so over data sequences).
> > >
> > > "Nonuniform" PAC/online are already "taken" as jargon terms, and have a distinct meaning (Example 2.7 of Bousquet et al., 2021, demonstrates a separation).
> > >
> > >
> > > > "on this sequence only countably many functions can be realized... need to build the enumeration in the online manner..."
> > >
> > > Thank you for clarifying your (very insightful) point.  As we understand it, your idea is to use the ordinal-SOA to produce a set of experts, analogous to the SOA-based experts in the agnostic part of the Daniely, Sabato, Ben-David, and Shalev-Shwartz (2015) paper, where the labeled training sets for the ordinal-SOA are based on "hallucinated" labelings of the examples.  We need only consider the (countably-many) possible labelings of finite sets of times, one of which coincides with the times (and labels) where the ordinal-SOA would make mistakes on the true labeled sequence.  This produces a countable family of experts, one of which is correct on the entire labeled sequence.  Then we can simply predict with the first expert in the list that can be induced from the observed data so far and yet hasn't been contradicted yet, and this eventually converges to an expert that never errs again in the rest of the sequence.
> > >
> > > This is a very nice idea.  If it's ok, we would like to include this idea in the final version of the paper (acknowledging an anonymous reviewer) in addition to our current technique.
> > >
> > > The solution in the submitted manuscript is not too far from this.  It produces a subset of the above experts algorithmically, in a branching fashion (incorporating the feedback), but rather than trying them one-by-one, it predicts with a weighted majority predictor, inspired by the work of Phil Long (2017).  We believe there is value to the weighted majority approach, though understanding this would require a quantitative comparison of mistake bounds to the simpler one-by-one algorithm.
> > >
> > > In any case, we hope you agree that, even if this analysis seems straightforward to a handful of experts who've spent a career thinking about these kinds of problems, the results have interesting aspects (the stark contrast of uniform vs universal mentioned above), which would be of interest to a broader set of researchers beyond these experts.

---

> > > > ### Comment · Reviewer_1b6p · 2024-11-26
> > > >
> > > > I don't agree with what was said about the infinite BL tree for the class of all constant functions. Of course, the previous notion was only defined for finite trees, so perhaps this is what caused misunderstanding. Let's agree that a BL tree is a complete infinitely branching tree, where nodes are labeled by points and edges are labeled by natural numbers (without repetitions). For all of this to make sense, we would like to say something as follows: the class $H$ shatters an infinite-depth BL tree $T$, if it shatters every path in $T$. Moreover, $H$ shatters an infinite path in $T$ if there exists a sequence of functions $h_1,h_2,\ldots$ in $H$ such that 1) for each $n$, $h_n$ disagrees with the labels on the $n$-length initial segment of the path AND 2) all natural $i$, $h_i$ and $h_{i+1}$ have the same labels on the first $i$ points in the path. To put it differently, there exists a function $h$ which disagrees with the labels on the path and $h$ is the limit of some sequence of functions from $H$.
> > > >
> > > > For the example of constant functions we can just forget about condition 2) because all the limits are already in the class. We should now see why this class does not shatter the tree given as the example (and in fact, any other tree of infinite depth). There is a path, in which we first go to the edge 1, then the edge 2, then the edge 3 and so on. Eventually, any constant function has to agree with something along this path. Indeed, this path gives us exactly the strategy of the learner to learn this class with a finite number of mistakes.
> > > >
> > > > It is not enough to just say that on each finite-depth subtree $H$ shatters all its paths. This argument just shows that this class cannot be learned uniformly. This is akin to the difference between having an infinite Littlestone tree and having unbounded Littlestone trees, as in the STOC paper.
> > > >
> > > > Let me say that I don't know for a fact if this notion already characterizes learnability in bandit setting. Perhaps some additional work is required. But this is exactly the kind of additional work I would like to see in this paper, before it can be published.
> > > >
> > > > As a side-note, I first saw Bousquet et al. paper less than 2 year ago, so my criticism is definitely not that of a some  seasoned gatekeeping expert who spent their whole career etc.

---

> > > > > ### Author Response · Authors · 2024-12-02
> > > > >
> > > > > We thank you for another fantastic suggestion. Your idea for a generalization of bandit Littlestone dimension is very interesting. We find that it indeed characterizes universal online learnability with bandit feedback:
> > > > >
> > > > > Let's call this type of tree a "universal bandit Littlestone tree" (UBL tree).
> > > > > 1. If there is an infinite UBL tree, then the class is clearly not learnable with bandit feedback (the adversary simply following the branch of the learner's predicted labels).
> > > > >
> > > > > 2. For the other direction, if the class is not learnable with bandit feedback, then our results imply there is an infinite Littlestone tree. From the latter fact, we can construct an infinite UBL tree: we will recursively modify the Littlestone tree to form a UBL tree, as follows. Starting from the root of the Littlestone tree, if the edge labels are $y$ and $y'$, we can make these the subtrees corresponding to labels $y'$ and $y$, respectively (i.e., just swap the labels), and for each other label $y''$ not among these two, we can take the subtree corresponding to label $y''$ equal the original $y$-child's subtree. Then recurse on each of these subtrees to generate the tree of the type you proposed. Since the original Littlestone subtree corresponding to label $y$ is shattered by the version space that labels the root as $y$, the resulting modified subtree will be shattered (in the proposed bandit sense) by functions labeling the root as $y$ (which differs from $y'$, the label of that edge in the modified tree), and similarly for each of the $y''$ subtrees, and analogously the subtree based on the Littlestone edge labeled $y'$ will be modified into the UBL tree, which is then shattered (in the proposed bandit sense) by the version space of functions labeling the root as $y'$ (which differs from $y$, the label of that edge in the modified tree). Hence this recursively modified tree is shattered in the sense you proposed.
> > > > >
> > > > > Perhaps more importantly, we can more-directly argue that there is a learning algorithm making finite mistakes when there is no infinite UBL tree. Consider the following Gale-Stewart game: on each round $t$, player $P_A$ proposes an $x_t \in \mathcal{X}$, then player $P_B$ proposes a $y_t \in \mathcal{Y}$; player $P_A$ wins if and only if there is a labeling $y_t^*$ of the infinite sequence of $x_t$'s which (1) avoids all the $y_t$ labels ($y_t^* \neq y_t$), and (2) is realizable by the concept class (i.e., all finite prefixes of the $(x_t,y_t^*)$ sequence are realizable). Clearly player $P_A$ has a winning strategy iff there is an infinite UBL tree. If player $P_B$ has a winning strategy, we can use it to produce an algorithm guaranteeing finite mistakes by running $P_B$'s winning strategy on the data sequence in ``conservative'' mode (only update $P_B$'s memory of the game with the mistake points in the sequence).  It remains only to argue that the game is determined: that is, for every concept class, one of the two players necessarily has a winning strategy.  For countable label spaces, we can use the above equivalence to infinite Littlestone trees to argue that, if there is no infinite UBL tree, then B has a winning strategy: since not having an infinite UBL tree implies there is no infinite Littlestone tree, we can use the countable set of experts based on the ordinal-SOA trained on finite subsequences of $x_t$'s with hallucinated labelings; just enumerate the countable set of experts (in a way such that the $i^{th}$ expert's training $x_t$'s are contained in the first $i$ times $t$), and let expert 1 propose $y_1$, expert 2 propose $y_2$, and so on, so that no infinite label sequence $y_t^*$ which avoids the $y_t$ values can be realizable.
> > > > >
> > > > > However, your idea has stimulated us to consider also a direct proof that the game is determined, based on the Borel Determinacy Theorem. This turns out to be non-trivial, since the set of winning sequences of $P_B$ is neither closed nor open (unlike the finitely-decidable games considered by Bousquet et al.), but it suffices to show it is a Borel set, and we are still discussing possible direct arguments that this holds. What is most interesting about this is that the argument based on this Gale-Stewart game does not depend on $\mathcal{Y}$ being countable. So if it is a determined game, this Gale-Stewart approach would provide a characterization of universal online learnability with general uncountable label spaces as well.  (We note that in the latter case, the equivalence to fully supervised learning no longer holds, even for constant functions)

---

### Meta-Review · Area_Chair_NbrK · 2024-12-09

**Recommendation:** Accept
**Confidence:** 2

**Metareview:**

To summarize all reviews and discussions:

(+) the results in the paper are new and makes a contrast with the uniform learning model;
(+) the reviewers appreciate the simplicity of the proof techniques;
(-) the results may be a bit small to pass the ALT threshold.
(-) discussions on the relationship with other papers, e.g. (Daniely et al, Multiclass Learnability and the ERM Principle) can be strengthened; specifically, the proof of Theorem 8 can benefit from the discussion of an alternative proof suggested by the reviewer and elaborated by the authors in their rebuttal.

Due to these considerations, this paper is on the borderline of acceptance.

**Paper Award:**

No